# Sulforaphane Ameliorates High-Fat-Diet-Induced Metabolic Abnormalities in Young and Middle-Aged Obese Male Mice

**DOI:** 10.3390/foods13071055

**Published:** 2024-03-29

**Authors:** Jing Luo, Hana Alkhalidy, Zhenquan Jia, Dongmin Liu

**Affiliations:** 1College of Food Science and Technology, Nanjing Agricultural University, Nanjing 210095, China; luojing@njau.edu.cn; 2Department of Human Nutrition, Foods, and Exercise, College of Agricultural and Life Sciences, Virginia Tech, Blacksburg, VA 24060, USA; haalkhalidy@just.edu.jo; 3Department of Nutrition and Food Technology, Faculty of Agriculture, Jordan University of Science and Technology, Irbid 22110, Jordan; 4Department of Biology, University of North Carolina at Greensboro, Greensboro, NC 27412, USA; z_jia@uncg.edu

**Keywords:** type 2 diabetes, insulin resistance, obesity, sulforaphane

## Abstract

Type 2 diabetes (T2D) is still a fast-growing health problem globally. It is evident that chronic insulin resistance (IR) and progressive loss of β-cell mass and function are key features of T2D etiology. Obesity is a leading pathogenic factor for developing IR. The aim of the present study was to determine whether sulforaphane (SFN), a natural compound derived from cruciferous vegetables, can prevent (prevention approach) or treat (treatment approach) obesity and IR in mouse models. We show that dietary intake of SFN (0.5 g/kg of HFD) for 20 weeks suppressed high-fat diet (HFD)-induced fat accumulation by 6.04% and improved insulin sensitivity by 23.66% in young male mice. Similarly, dietary provision of SFN (0.25 g/kg) significantly improved blood lipid profile, glucose tolerance, and insulin sensitivity of the middle-aged male mice while it had little effects on body composition as compared with the HFD group. In the treatment study, oral administration of SFN (40 mg/kg) induced weight loss and improved insulin sensitivity and plasma lipid profile in the diet-induced-obesity (DIO) male mice. In all three studies, the metabolic effects of SFN administration were not associated with changes in food intake. In vitro, SFN increased glucose uptake in C2C12 myotubes and increased fatty acid and pyruvate oxidation in primary human skeletal muscle cells. Our results suggest that SFN may be a naturally occurring insulin-sensitizing agent that is capable of improving the metabolic processes in HFD-induced obesity and IR and thereby may be a promising compound for T2D prevention.

## 1. Introduction

The diabetes mellitus pandemic continues to be a growing public health issue as 37 million Americans [1] and 537 million people worldwide have been diagnosed with diabetes [2]; this number has nearly quadrupled since 1980 [3]. If such a trend persists, more than 783 million people will have diabetes by 2045 [2]. The most common form of diabetes is type 2 diabetes (T2D), accounting for roughly 90% of all diabetic patients [4]. The pathogenesis of T2D includes a progression from chronic insulin resistance and the loss of β-cell mass and function to a resultant inadequate capacity of the β-cells to secrete insulin in sufficient quantities to compensate for the decrease in insulin sensitivity [5,6,7,8,9]. Despite novel therapeutic drugs and approaches to diabetes treatment that have considerably improved glucose control and survival rate in patients with diabetes, the disease is still poorly managed for at least half of T2D patients [10]. Because of this, the pursuit of novel bioactive molecules that can be used as complementary or adjuvant therapy for the pathogenesis of T2D, and its secondary conditions, is of great interest.

Sulforaphane (SFN) is an isothiocyanate synthesized from the glucosinolate glucoraphanin (structures of SFN and glucoraphanin as shown in Figure 1), which is naturally present in cruciferous vegetables, including broccoli, brussels sprouts, cabbage, and cauliflower. Myrosinase is an intracellular thioglucosidase, found in the normal flora of the gut, and it can catalyze glucoraphanin into sulforaphane. Several human studies have demonstrated the profound health benefits of SFN in its ability to decrease the incidence of various types of cancer [11,12,13] and diabetes [14,15,16]. More specifically, SFN elicits a cellular antioxidative effect, which is mediated by disrupting Nrf2-Keamp1 association, allowing Nrf2 to translocate to the nucleus where it induces the expression of antioxidant genes [17]. In addition, SFN has been shown to be a beneficial intervention for vascular complications associated with type 1 diabetes [15], and to elicit a protective effect in the aortas of mice with T2D by diminishing biomarkers of fibrosis, inflammation, and oxidative stress [18,19]. Likewise, we recently demonstrated that SFN inhibits TNF-α-induced vascular endothelial inflammation, in vivo and in vitro, by impeding the adhesion of monocytes to vascular endothelial cells while suppressing NF-κB signaling, which would otherwise result in augmented inflammation [20]. As such, it is of particular interest to determine whether SFN can extend its anti-inflammatory properties to elicit anti-diabetics effect in the context of attenuated blood glucose and improved insulin sensitivity. Thus, we tested in the present study whether SFN can work as a viable dietary intervention to ameliorate hyperglycemic conditions in obese, insulin-resistant, and glucose-intolerant mice, and to test this we used young and middle-aged male mice. Herein, we report for the first time, to our knowledge, that dietary SFN at an achievable dosage exerts potent effects on ameliorating HFD-induced adiposity and hyperglycemia. SFN treatment effectively lowered blood triglyceride and cholesterol levels in obese male mice. Our findings indicate that SFN could be used as a dietary supplement to prevent, and perhaps aid in the treatment of obesity-related metabolic disease.

## 2. Materials and Methods

### 2.1. Animal Studies

Male C57BL/6J mice (NCI, Frederick, MD, USA) were housed in a vivarium maintained at ambient temperature (22–25 °C) and on a 12 h light/dark cycle. The animals had *ad libitum* access to food and water. Following a one-week period of environmental acclimation, the following three animal studies were performed. All protocols for animal experiments were approved by the Institutional Animal Care and Use Committee at Virginia Tech and the approval code is IACUC-12-201.

#### 2.1.1. Prevention Studies in Young Mice

To assess the long-term dietary effects of sulforaphane in the development of diet-induced adiposity, six-week-old male mice (C57BL/6, NCI/NIH) with balanced body weight (BW), body composition, and non-fasting (NFBG) and fasting blood glucose (FBG) were divided into three groups (*n* = 16/group): control group (standard chow diet with 10% calories from fat, STD), high-fat diet group (with 60% calories from lard, HFD), and HFD supplemented with 0.5 g/kg SFN group (HFD + SFN). This study was maintained for 18 weeks. Body weight and food intake were measured weekly. NFBG and FBG were measured every other week. Body composition and blood pressure were measured at weeks 9 and 13. Glucose tolerance test (GTT) and insulin tolerance test (ITT) were performed at weeks 12 and 18. For measuring FBG, mice were fasted for 12 h, and blood glucose in the tail vein was measured using a glucometer (Kroger, Cincinnati, OH, USA).

Body composition was evaluated by using an LF-90 instrument (Bruker Optics Inc., Billerica, MA, USA), which is based on time-domain nuclear magnetic resonance technology. This allows for an in vivo measurement of lean body mass, fat body mass, and body fluid without the use of anesthesia. For the intraperitoneal glucose tolerance test (GTT), mice were fasted for 12 h and subsequently injected with a 1 g/kg BW bolus of glucose. Blood glucose was measured at 0, 15, 30, 60, and 120 min time points. For the intraperitoneal insulin tolerance test (ITT), mice were fasted for 4 h and then injected with 0.75 units/kg BW insulin. Blood glucose was measured at 0, 15, 30, 60, and 120 min time points. The area under the curve (AUC) was calculated by using the trapezoidal rule. Blood pressure was measured using the Kent CODA 2 blood pressure system as previously described [21] with minor alterations. Briefly, mice were calmed by a warming platform connected to the Kent CODA 2 series computerized noninvasive blood pressure system. The systolic and diastolic blood pressure measurements were recorded for 20 cycles in each mouse, and the highest and lowest values were excluded. All procedures were performed gently by the same person to avoid variations.

#### 2.1.2. Prevention Studies in Middle-Aged Mice

Eight-month-old C57BL/6 male mice were divided into 3 groups (*n* = 8/group) with blood glucose and BW balanced. For 7 weeks, the animals were fed a standard chow diet (STD), with 10% of the calories from fat; a HFD (Research Diets Inc., New Brunswick, NJ, USA), with 60% of calories from fat; or a HFD supplemented with SFN (0.25 g/kg diet) (HFD+SFN). Body weight and food intake were recorded on a weekly basis for the duration of the study. After 7 weeks of SFN supplementation, FBG, body composition, GTT, and ITT were measured as instructed above. At the end of the study, mice were euthanized following an overnight fast, and blood samples were immediately collected. Plasma insulin was measured using an ultrasensitive mouse/rat insulin ELISA kit (Mercodia, Winston Salem, NC, USA), and plasma cholesterol and triglycerides were assayed with a Pointer 180 Analyzer (Pointe Scientific, Canton, MI, USA).

#### 2.1.3. Treatment Studies in Middle-Aged Obese Mice

For this study, 8-month-old male mice (NCI, NIH) were fed a HFD for 8 weeks until mice became obese with an average body weight of 50 g. Mice were divided into 2 groups (*n* = 15 mice/group) with similar body weights, body compositions, and blood glucose. Both groups were fed an HFD; one group was administered with 40 mg/kg SFN via oral gavage, and the other was gavaged with an equivalent volume of the vehicle (2% methylcellulose) for 30 days. Body weights were recorded every three days and food consumption was measured every week. NFBG and FBG were measured bi-weekly as described in the dietary intervention study.

After 30 days of SFN administration, GTT, ITT, and body composition were evaluated as described above. Body temperature was measured using a thermometer probe placed in the rectum at 2.5 cm depth. Energy expenditure was evaluated using the indirect calorimetry system by assessing oxygen (O_2_) consumption and carbon dioxide (CO_2_) output. Regarding this, mice were individually placed in a TSE LabMaster Calorimetry System cage (Columbus Instruments, Columbus, OH, USA). Following acclimation for 24 h, mice were hooked up to the TSE LabMaster System, in a closed environment, which allowed for metabolic sampling. Mice had free access to food and water for the duration of this study. Oxygen consumption, CO_2_ content, and locomotor activity were recorded for 48 h. All data were normalized using the lean mass of body weight. Following these procedures, the mice were fasted overnight and euthanized, and samples of blood were collected to be assayed for cholesterol and triglyceride contents. Total cholesterol and triglycerides in the plasma were measured by using lipid reagents (Teco Diagnostics, Anaheim, CA, USA). Briefly, 1.0 mL of the respective reagents were added to tubes and incubated at 37 °C for 4 min. An amount of 10 μL of the sample was added to this reagent, mixed, and incubated at 37 °C for 10 min and 5 min for measuring cholesterol and triglycerides, respectively. Following incubation, the absorbance was read with a spectrophotometer at 520 nm. Absorbance was normalized to the standard, multiplied by the concentration of the standard, and expressed as mg/dL.

Total lipids were extracted from fecal samples using the Folch method [22,23]. Briefly, fecal samples were homogenized in chloroform/methanol (2:1) solution with a final dilution 20 times the volume of the sample. The homogenates were then filtered and the crude extracts were washed with 0.2 of its volume of PBS. The solutions were then allowed to separate into two phases and the upper layer was removed by siphoning. The extracts of fecal samples were dried and individually weighed, and then total cholesterol and triglycerides were determined by chemistry reagent kits as instructed (Teco Diagnostics, Anaheim, CA, USA).

### 2.2. Cell Culture

C2C12 myoblasts (American Type Culture Collection, Manassas, VA, USA) and primary human muscle cells, obtained from subjects who provided written informed consent under a protocol, approved by Virginia Tech Institutional Review Board [24], were cultured for the following tests:

#### 2.2.1. Glucose Uptake Assay

C2C12 mouse skeletal muscle cells were cultured to a confluence of 80–90% in Dulbecco’s modified Eagle’s medium (DMEM) containing 10% heat-inactivated fetal bovine serum (FBS) and 1% penicillin streptomycin (P/S). Cells were then differentiated into myotubes in DMEM with 2% horse serum (HS) and 1% P/S. All experiments were conducted between days 5 and 7 of differentiation, at which time the formation of myotubes reached the maximum. For testing if SFN has a direct effect on glucose uptake, differentiated myotubes were treated with SFN (0, 1 nM, 10 nM, 50 nM, 100 nM, 1 μM) in DMEM for 1 h at 37 °C. Insulin treatment (100 nM) was included as a positive control. Glucose uptake was measured using a fluorescence assay kit according to the manufacturer’s protocol (Cayman Chemicals, Ann Arbor, MI, USA). Briefly, the culture medium was replaced with 15 μL/mL of fluorescent glucose analog2-(N-(7-Nitrobenz-2-oxa-1,3-diaz-4-yl) Amino)-2-Deoxyglucose (2-NBDG) in PBS at 37 °C for 10 min. The plate was centrifuged (400 rcf for 5 min at 25 °C), the supernatant was aspirated, and 200 μL of cell-based assay buffer was added to each well for a secondary centrifuge step. The supernatant was aspirated and replaced with 100 μL of cell-based assay buffer; the plate was immediately analyzed by determining its fluorescence (excitation/emission: 485/528 nm) with a plate reader (BioTek Synergy 2 plate reader, Gen5 Microplate Data Collection and Analysis Software version 2.0, Winooski, VT, USA). The glucose uptake stimulated by different doses of SFN was then calculated as the fold change compared to the control group.

#### 2.2.2. Fatty Acid and Pyruvate Oxidation

Primary human muscle cells were cultured as described [25]. Fatty acid and pyruvate oxidation were assessed by measuring and summing ^14^CO_2_ production and ^14^C-labeled acid-soluble metabolites from the oxidation of [1-^14^C] palmitic acid (Perkin Elmer, Waltham, MA, USA) and [U-^14^C] pyruvate (American Radiolabeled Chemicals, St. Louis, MO), as described previously [26]. Briefly, SFN (0, 10 nM, 100 nM, 1 μM, and 2 μM) was added to the culture medium. Then, 1 mL of the incubation medium was transferred to a glass scintillation vial, containing 1 mL of 1 M H_2_SO_4_, and a microcentrifuge tube, which contained benzenthonium hydroxide. The ^14^CO_2_ that was generated was trapped in a small microcentrifuge tube with benzenthonium hydroxide. This tube was placed in the scintillation vial and measured. The absolute quantities of palmitic acid and pyruvate oxidized were determined by measuring the radioactivity in vials containing the trapped ^14^CO_2_ and ^14^C-water-soluble metabolites. A ratio of substrate to the oxidized products was utilized to examine fatty acid and pyruvate oxidation differences in primary human skeletal muscle cells.

### 2.3. Data Analysis

All data were analyzed by one-way ANOVA using SPSS 25 statistics software and are expressed as Mean ± SEM. Differences between group means were compared by Tukey’s HSD Test with *p*-value < 0.05 considered significant.

## 3. Results

### 3.1. Long-Term Dietary Supplementation of SFN Ameliorated HFD-Induced Metabolic Abnormalities in Young Male Mice

To observe the effects of dietary supplementation of SFN on the development of adiposity, we fed 6-week-old male mice a HFD supplemented with 0.5 g/kg SFN for 20 weeks. Starting from week 2, the body weight of the HFD group significantly diverged from that of the STD and HFD+SFN groups (Figure 2A). The HFD group weighed 26.60 ± 0.54 g, whereas the HFD+SFN group weighed 24.64 ± 0.50 g, which is comparable to the STD group (23.12 ± 0.43 g). At week 20, the HFD group weighed 47.98 ± 0.86 g, while the SFN supplement significantly suppressed body weight gain (44.76 ± 1.20 g) as compared with the HFD group. Interestingly, the HFD+SFN group consumed more food than the other two groups in the first week, which may suggest an attractive flavor of SFN to the mice. Nonetheless, the food intakes of the HFD and HFD+SFN groups were all the same from week 2 throughout the study (Appendix A).

To investigate the effects of SFN on glycemic control in young male mice, we measured non-fasting (NFBG) and fasting blood glucose (FBG) every other week. The NFBG of the three groups were similar throughout the study (Appendix A), suggesting an intact glycemic control in young male mice. Interestingly, however, the FBG levels of the HFD group were constantly the highest among the three. We observed that dietary supplementation with SFN significantly blunted HFD-induced hyperglycemia at weeks 2, 6, and 8 (Figure 2B).

We measured body composition and blood pressure in week 9. To avoid stress to the mice, we evenly divided each group into two subgroups for these two measurements. Dietary intervention with SFN significantly suppressed HFD-induced adiposity by 6.04%, whereas the percentage of lean mass was elevated in the SFN group as compared to the HFD group by 5.96% (Figure 2C). As shown in Figure 2D, dietary supplementation with SFN lowered diastolic and systolic blood pressure in the HFD-induced obese mice as compared to the HFD group, suggesting SFN intervention may help in attenuating hypertension.

To investigate the effects of SFN supplementation on glucose homeostasis, we performed GTT and ITT in week 18. Dietary intervention with SFN did not affect glucose tolerance status in the HFD-induced obese mice (Appendix A). However, as shown in Figure 2E of the ITT result, the SFN supplement strongly improved insulin sensitivity in the hyperglycemic obese mice as compared with the HFD group by 23.66% (Figure 2F).

### 3.2. Dietary Intake of SFN Enhanced Glucose Homeostasis and Lipid Profile in Middle-Aged Obese Mice

Since we observed the glycemic-controlling effects of long-term dietary SFN supplementation in young mice, we further tested whether dietary SFN also exerts beneficial metabolic effects in middle-aged mice as T2D typically occurs after middle and older ages in humans. Mice were divided into three groups as indicated in the Materials and Methods and SFN supplementation was mixed in the HFD (0.25 g/kg diet). As shown in Figure 3A, mice fed the HFD had heavier body weight and developed obesity, which was determined by measuring body weight and the relative percentage of body fat and lean muscle mass (Figure 3B,C), and this effect was not negated by SFN. Short-term SFN supplementation marginally lowered (*p* = 0.106) HFD-induced hyperglycemia (Figure 3D). We also measured plasma insulin levels following an overnight fast. As shown in Figure 3E, the fasting plasma insulin level of the SFN group was approximately 40% lower than that of the HFD group, suggesting short-term intervention with SFN may ameliorate hyperinsulinemia condition in diabetic mice. Short-term SFN treatment significantly lowered fasting blood cholesterol by 25.5% (Figure 3F) and blood HDL by 20.3% as compared with the HFD group (Figure 3G) levels. The blood TG levels were lower in the SFN group but not significant when compared with the HFD-alone group (Figure 3H).

### 3.3. Dietary Supplementation with SFN Promoted Insulin Sensitivity and Ameliorated Hyperglycemia Induced by HFD

As a part of our attempt to determine whether SFN can delay the etiology of T2D, we performed GTT and ITT after 30 days of treatment. Two hours after a glucose challenge, mice fed an HFD supplemented with SFN had significantly lower blood glucose than the animals fed an HFD alone (Figure 4A). It is worth noting that these levels are statistically similar to the animals being fed an STD. The SFN supplement significantly ameliorated hyperglycemia induced by HFD as indicated by a 21.4% decrease in the AUC of the GTT result (Figure 4B). As was expected, HFD feeding impaired insulin sensitivity (Figure 4C), but dietary supplementation with SFN significantly promoted insulin sensitivity as shown by a 31.9% decrease in the AUC of the ITT result as compared with the HFD group (Figure 4D).

### 3.4. Treatment with SFN Promotes Weight Loss in Diet-Induced Obese Mice

To determine whether SFN may potentially act as an anti-diabetic agent and improve the health and metabolic function of obese mice, 8-month-old diet-induced obese (DIO) male mice were switched from an HFD to an STD and orally fed with either the vehicle (2% cellulose) or 40 mg/kg BW SFN for 30 days. SFN did not affect food intake (Appendix A); however, it significantly enhanced weight loss starting from 6 days of SFN administration and lasted until day 24 of SFN administration (Figure 5A). SFN treatment did not change body fat and lean mass (Figure 5B) at the end of the study, nor did it affect blood glucose profile (Appendix A). Further, SFN treatment did not change physical activity, either during the day or night (Figure 5C), or the average energy expenditure as measured via indirect calorimetry (Figure 5D). To investigate the effects of SFN on glycemic control, we performed a GTT and found that SFN treatment did not affect glucose tolerance in the old DIO male mice (Appendix A). However, SFN significantly improved insulin sensitivity, which is denoted by the significantly lower glucose levels at 30 and 60 min after insulin injection (Figure 5E) and improved insulin sensitivity by 14.6% as compared with the control (Figure 5F).

### 3.5. Treatment with SFN Promoted Lipid Profile in DIO Male Mice and Enhanced Energy Oxidation In Vitro

Oral gavage with SFN significantly lowered TG and TC concentrations in circulation by 18.8% and 17.8%, respectively (Figure 6A). Interestingly, the fecal TG and TC levels tended to be higher in the SFN group (Figure 6B), suggesting that treatment with SFN may suppress intestinal fat absorption, which needs further investigation.

To test our speculation that SFN intervention improves insulin sensitivity via enhancing glucose metabolism in peripheral tissues, we conducted a cell-based glucose uptake assay. To that end, C2C12 cells were first induced to differentiate into myotubes as described [27]. Then, C2C12 myotubes were cultured with or without SFN for 60 min. As shown in Figure 6C, SFN promoted glucose uptake in C2C12 myotubes with 10 nM increasing by 9% and 100 nM increasing by 12% over the control, whereas 100 nM insulin increased glucose uptake by 14%. SFN treatment at 1 nM, 50 nM, and 1 μM can also marginally increase glucose uptake in C2C12 myotubes.

To further evaluate whether SFN affects energy metabolism in muscle cells, primary human skeletal muscle cells were incubated with palmitic acid in the presence of SFN or the vehicle followed by measuring CO_2_ production. SFN treatment increased the production of CO_2_ at 100 nM, 1 μM, and 2 μM, respectively, with 1 μM showing statistical significance (Figure 6D). Consistently, SFN also enhanced pyruvate oxidation in a dose-dependent manner in primary human skeletal muscle cells, with 1 μM SFN increasing pyruvate oxidation by approximately 49% as determined by measuring CO_2_ production from pyruvate oxidation (Figure 6E).

## 4. Discussion

SFN, a bioactive compound derived from cruciferous vegetables, has drawn great interest for its health-promoting effects. It has been extensively investigated for its anticancer, anti-inflammation, and antioxidant capacities [28,29,30]. In the present study, we found that dietary intervention with SFN had metabolic benefits in young, middle-aged, and DIO male mice. Specifically, long-term dietary intake of SFN promoted metabolic homeostasis by lowering body fat and enhancing insulin sensitivity in young mice fed an HFD. Further, SFN intervention, with a lower dosage, protected middle-aged male mice from HFD-induced glucose intolerance, insulin resistance, and hyperlipidemia. Interestingly, SFN treatment preserved these metabolic benefits in DIO male mice even when they were switched to a standard chow diet. None of these beneficial effects is dependent on food intake. We further demonstrated that SFN treatment increased glucose uptake in C2C12 myotubes and boosted energy metabolism in primary human skeletal muscle cells. As such, SFN could be a naturally occurring compound that promotes glucose homeostasis and energy metabolism as both a preventative and a therapeutic agent.

Insulin resistance and obesity are always associated with impaired energy metabolism in peripheral tissues. As such, HFDs are widely used to stimulate the development of insulin resistance, glucose intolerance, hyperinsulinemia, and hyperlipidemia [31,32,33]. Several studies have noted the implication that sustained hyperlipidemia contributes to β-cell apoptosis and dysfunction, which is largely responsible for diminished glycemic control and overt T2D development [34,35,36]. To this end, we designed two independent animal studies with long- and short-term HFD feeding. Only in the short-term study does SFN intervention exert a profound capability in glycemic control. By the end of the long-term intervention study, the beneficial effects of the SFN supplement on glycemic control were blunted as the GTT results showed no significant difference. Considering this together with the dramatic improvement in insulin sensitivity, we speculated this may potentially be due to the highly interrupted function of pancreatic islets to release insulin after long-term HFD feeding. However, we indeed observed that SFN ameliorated hyperglycemia at an early stage of the experiment and later the effects faded away. Consistent with our findings, administration of SFN (0.56 g/kg) for 6 weeks significantly improved glucose tolerance and insulin sensitivity in male mice with non-alcoholic fatty liver disease [37]. Therefore, it is possible that the promising effects of the SFN supplement would peak in a short period of time. Another possibility is that the beneficial effects of SFN depend on the dosage and the age group of subjects, which hence needs further investigations.

The dosage in the present study varied from 0.25 g/kg diet to 40 mg/kg body weight. In the long-term intervention study, 0.5 g/kg SFN showed quite promising benefits on body weight control and insulin sensitivity. Interestingly, however, the glucose tolerance was not affected in the young male mice. We speculated that the metabolic flexibility was not impaired in those young mice and the pancreatic β-cells remained intact. In the short-term intervention study, glucose tolerance and insulin sensitivity were both significantly enhanced with 0.25 g/kg SFN, which would be translated to a human equivalent dose of 100 mg for an adult weighing 60 kg [38,39]. Our findings indicate that SFN exerts metabolism-boosting effects at an achievable concentration as a supplement.

Moreover, sex heterogeneity in energy metabolism could be an influencing factor. Dietary supplementation with dried broccoli sprouts (BrSp, a good source of SFN) (300 mg/kg) ameliorated hypertension and improved glucose tolerance in male Long–Evans rats and suppressed visceral fat accumulation in female rats [40]. Pretreatment with SFN (5 mg/kg) significantly reduced the expression of oxidative stress and mitochondrial dysfunction markers in 12-day-old male Wistar rats [41]. Intraperitoneal injection of SFN (10 mg/kg) for 30 days reduced HFD-induced body weight gain in female mice [42]. The expression of UCP-1, a key regulator in white adipose tissue browning, was higher in SFN-treated female mice and so were the mitochondrial biogenesis genes [42], suggesting a promising role of SFN in energy metabolism in females. We did not observe huge changes in body composition in male mice; however, SFN treatment boosted energy oxidation in skeletal muscle cells, suggesting that SFN may promote weight loss through enhancing fatty acid oxidation and cellular energy expenditure, which remains to be examined.

Oral treatment with SFN significantly enhanced the translocation of GLUT4 in the gastrocnemius muscle of young male mice [43], which could partially explain our results that SFN treatment stimulated glucose uptake in skeletal muscle cells. Thus, it is urgent to unravel the potential cellular signaling pathways regulating the beneficial effects of sulforaphane.

Broccoli seed extracts preserved intestinal barrier function and changed the gut microbiota structure in dextran sulfate sodium-induced colitis male mice [44]. In the present study, weight loss and changes in the lipid profile were independent of SFN, and it is possible that these changes may have been mediated by a change in the intestinal flora or digestion and absorption capabilities. Because weight loss is, very broadly speaking, a direct result of a positive shift in energy balance, it is also possible that unregulated physical activity may be affected by SFN. Uncovering the underlying mechanisms and potential determinants to determine a favorable dosage and duration of SFN intervention may help in recommending naturally occurring anti-diabetic and anti-obesity agents.

## 5. Conclusions

In summary, 20 weeks of 0.5 g/kg SFN feeding significantly suppressed HFD-induced body fat accumulation by 6.04% and improved insulin sensitivity by 23.66% as compared to HFD-fed young male mice. Further, 7 weeks of 0.25 g/kg SFN significantly decreased blood cholesterol and HDL levels in HFD-fed middle-aged male mice. Glucose tolerance and insulin sensitivity were improved by SFN supplementation. We propose that SFN treatment promoting glucose uptake, FAO, and pyruvate oxidation in skeletal muscle cells may represent a core mechanism contributing to enhanced metabolic abnormalities induced by HFD; hence, SFN may act as an insulin-sensitizing agent. These data show that SFN could be used as a dietary supplement to prevent, and perhaps aid in the treatment of obesity-related metabolic disease.

## Figures and Tables

**Figure 1 foods-13-01055-f001:**
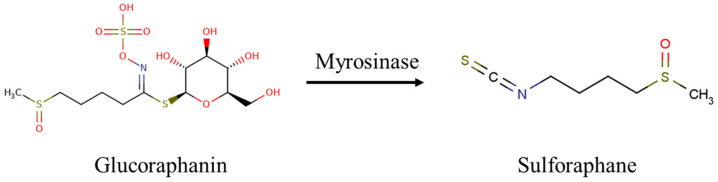
The chemical structures of glucoraphanin and sulforaphane.

**Figure 2 foods-13-01055-f002:**
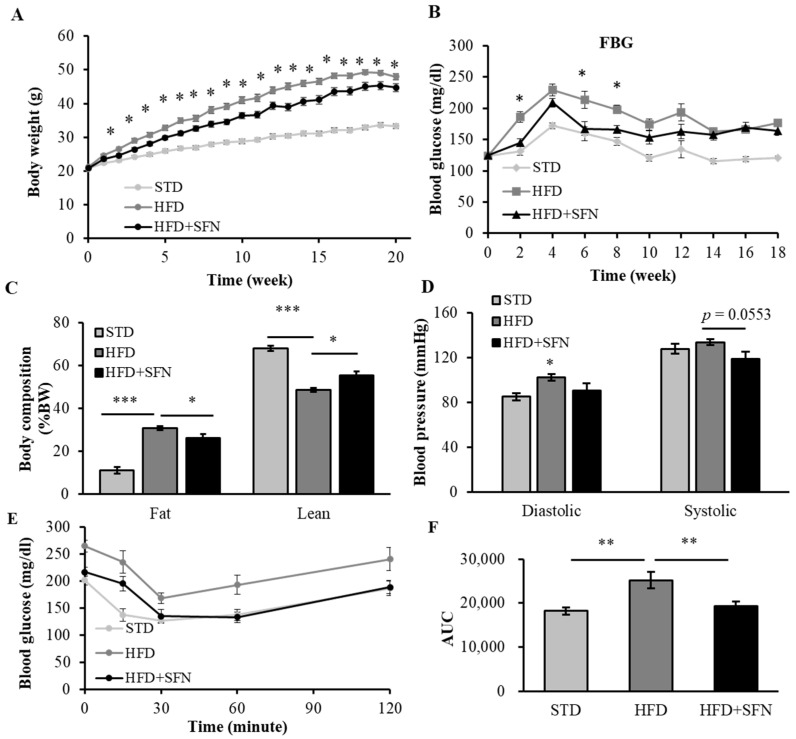
Long-term dietary supplementation with SFN suppressed HFD-induced adiposity and improved insulin sensitivity and hypertension status in young mice: (**A**) While HFD feeding results in the development of obesity showed that dietary SFN supplementation had little effect on body weight in young mice, (**B**) SFN supplement significantly lowered FBG at weeks 2, 6, and 8. (**C**) HFD feeding increased body fat and decreased lean mass in week 9, while SFN supplement significantly lowered body fat accumulation and increased lean mass as compared to the HFD group. (**D**) SFN supplement ameliorated hypertension as compared to HFD at week 9. (**E**) SFN supplement significantly improved insulin sensitivity as compared to HFD group at week 18. (**F**) AUC of ITT results. Data are shown as Mean ± SEM (*n* = 16) (* *p* < 0.05, ** *p* < 0.01, *** *p* < 0.001). STD: standard chow diet; HFD: high-fat diet; HFD+SFN: HFD supplemented with sulforaphane (0.5 g/kg diet).

**Figure 3 foods-13-01055-f003:**
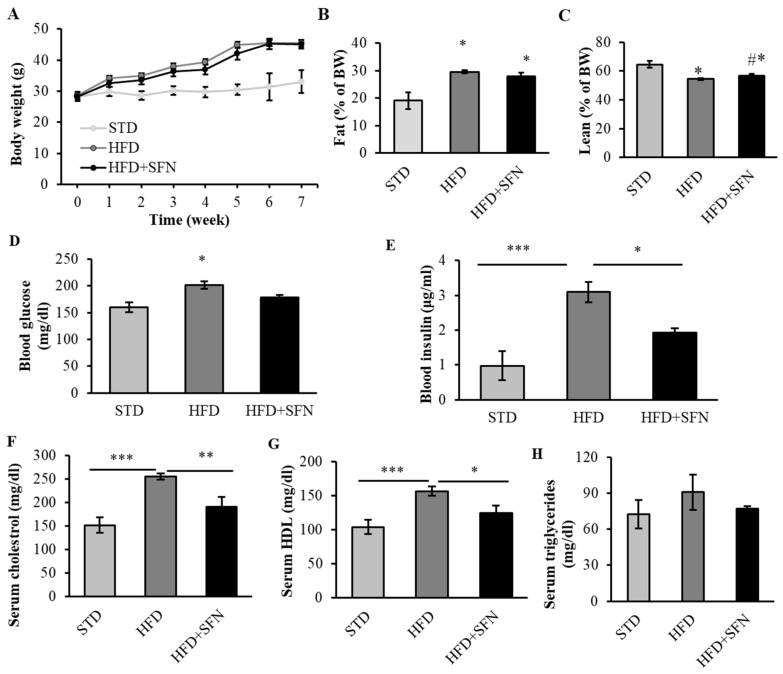
Short-term dietary intake of SFN enhanced glucose homeostasis and lipid profile in HFD-fed obese mice. Short-term HFD supplemented with 0.25 g/kg SFN had little effect on body weight in middle-aged male mice (**A**). Body composition was measured following 7 weeks of HFD and fat mass was significantly higher in both HFD feeding groups (**B**). SFN supplement increased lean mass as compared to HFD group after 7 weeks of HFD feeding (**C**) (* *p* < 0.05 vs. STD, # *p* < 0.05 vs. HFD). Fasting blood glucose was higher in HFD group and SFN slightly lowered blood glucose but not statistically significantly (**D**). SFN intervention significantly decreased fasting insulin concentration after 7 weeks (**E**). Dietary supplementation with SFN significantly decreased serum cholesterol (**F**) and HDL (**G**), and slightly decreased triglycerides (**H**) after 7 weeks of treatment. Data are displayed as Mean ± SEM (*n* = 8) (* *p* < 0.05, ** *p* < 0.01, *** *p* < 0.001). STD: standard chow diet; HFD: high-fat diet; HFD+SFN: HFD supplemented with SFN (0.25 g/kg diet).

**Figure 4 foods-13-01055-f004:**
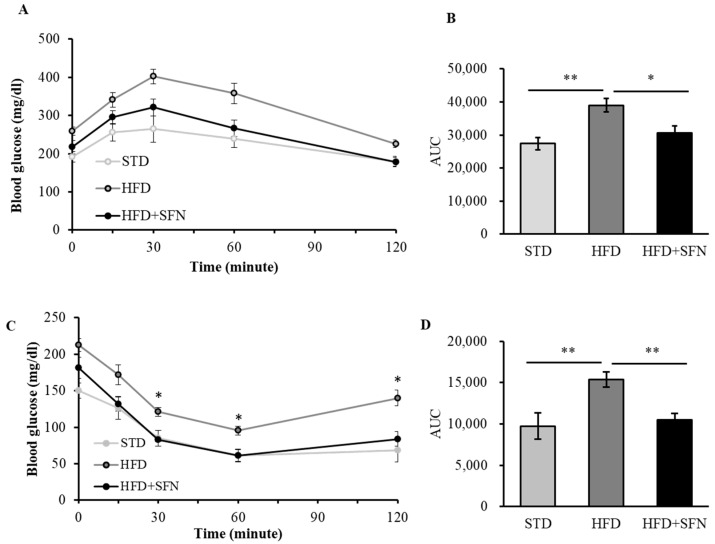
Short-term dietary supplementation of SFN promoted insulin sensitivity and ameliorated hyperglycemia induced by HFD. Blood glucose levels were measured after a glucose challenge (**A**) and AUC was calculated (**B**). SFN supplement significantly improved glucose tolerance as compared to HFD group after 7 weeks of feeding. Blood glucose levels were measured after an insulin challenge (**C**) and AUC was calculated (**D**). SFN supplement significantly improved insulin sensitivity as compared to HFD group after 7 weeks of feeding. Data are displayed as Mean ± SEM (*n* = 8) (* *p* < 0.05, ** *p* < 0.01). STD: standard chow diet; HFD: high-fat diet; HFD+SFN: HFD supplemented with SFN (0.25 g/kg diet).

**Figure 5 foods-13-01055-f005:**
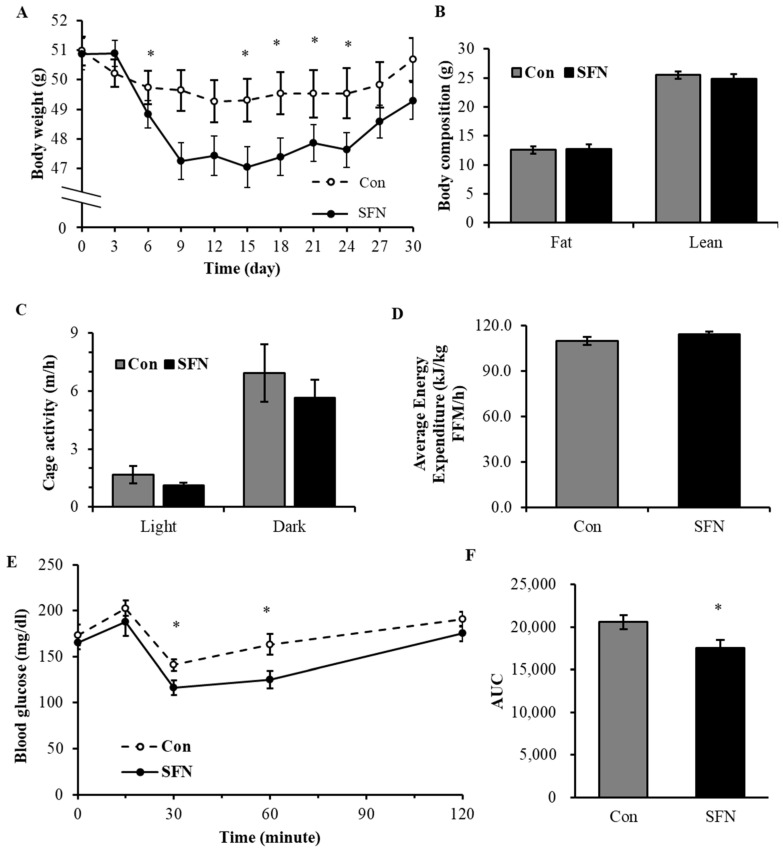
Treatment with SFN promotes weight loss and insulin sensitivity in diet-induced obese mice. SFN enhanced body weight loss (**A**). There was no effect on body composition (**B**), cage activity (**C**), or the average energy expenditure (**D**). SFN treatment improved insulin sensitivity after 30 days (**E**,**F**). Data are displayed as Mean ± SEM (*n* = 9–15) (* *p* < 0.05). Con: control group (oral gavage with vehicle); SFN: SFN treatment group (oral gavage with 40 mg/kg SFN).

**Figure 6 foods-13-01055-f006:**
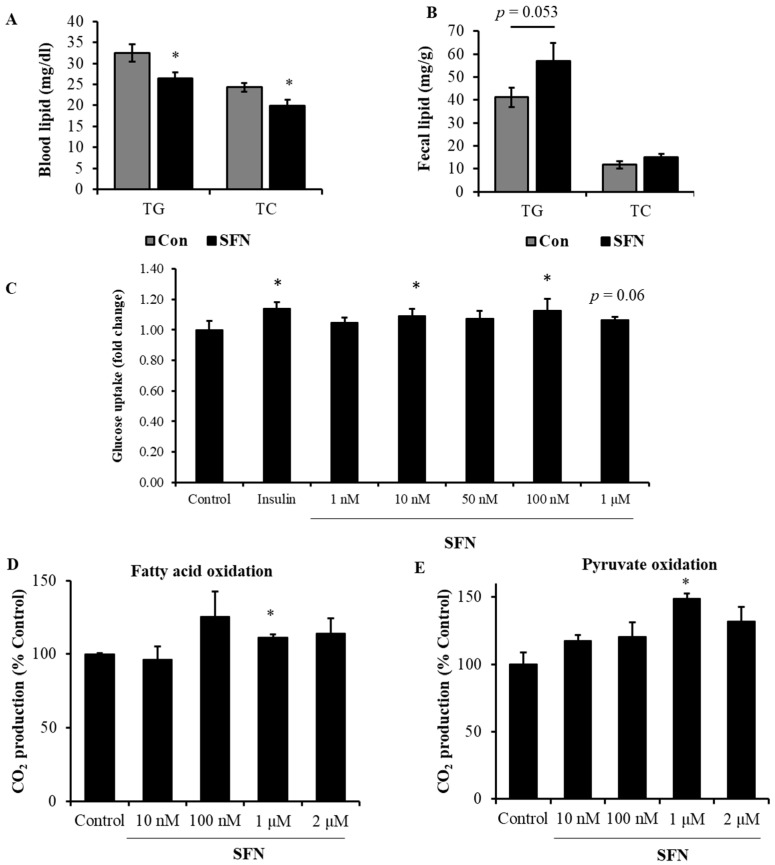
Treatment with SFN promoted lipid profile in DIO mice and enhanced energy oxidation in vitro. Circulating triglycerides and total cholesterol were lower in SFN treatment group (**A**). Fecal lipid profile was slightly higher in SFN treatment group (**B**). Data are displayed as Mean ± SEM (*n* = 7–9). HF: high-fat diet + vehicle; HF+SFN: HF diet + 40 mg/kg sulforaphane in vehicle. (**C**) SFN treatment at 10 nM and 100 nM significantly increased glucose uptake in C2C12 differentiated myotubes. SFN treatment at 1 μM significantly increased fatty acid oxidation (**D**) and pyruvate oxidation (**E**) in primary human skeletal muscle cells. Data are displayed as Mean ± SEM (*n* = 3) (* *p* < 0.05 vs. control).

## Data Availability

All the data and materials are available.

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
