# Peer review of "Sulforaphane Ameliorates High-Fat-Diet-Induced Metabolic Abnormalities in Young and Middle-Aged Obese Male Mice"

_foods, 2024, doi:10.3390/foods13071055_

Round 1

Reviewer 1 Report

Comments and Suggestions for Authors

In the submitted manuscript (foods-2925276), the authors comprehensively analyzed the effects of the bioactive plant component sulforaphane on metabolic abnormalities in young and middle-aged mice induced by an inadequate (high-fat) diet supported by experiments on cell cultures. Numerous positive effects of supplementation are observed and discussed, with the general conclusion that the investigated compound (e.g., in the form of a dietary supplement) would show a promising effect on preventing, perhaps even treating, metabolic syndrome/disease.

Simply put, it is a very good study, meaningfully and clearly conceived and executed (kudos to 3 different in vivo studies), with very concisely (perhaps too much) presented and discussed findings and implications for further research.

I have no essential (critical) objections, but I have more questions and suggestions about finishing the existing text ("to a high gloss").

1. Please introduce the meaning of the acronyms used after the FIRST mention (both in the summary/abstract and in the text of the paper itself, starting with the Introductory part), e.g. (abstract) HFD (OK, you can tell from the title), but DIO (no clue for readers).

2. Check whether all Latin words/expressions (via, in vitro, etc.) are in italics throughout the text. I believe they are not!

3. The tested bioactive compound (sulforaphane) is not well known to the general readership (except perhaps as a neoplastic agent). Therefore, it would be helpful to include its chemical structure (possibly with the formula of glucoraphanin, its glucosinolate precursor) in the text, as a separate figure, or as part of some figure with the study results.

4. Some parts of the text with experimental protocols must be completed, specifically section 2.2.1. and 2.2.2. In the specified order, the fluorescences were measured, and/or the samples were placed in the scintillation counter, and then/so what? Please complete, i.e., clarify!

5. Put everywhere with the results of the study, along with claims that the observed differences (for the measured parameters) are significant (or a "strong" effect was obtained), at what level (give P value!), as well as by how much (is a specific parameter higher/lower) in percentages. Namely, the results are (mostly) presented through histograms, from which it takes work to see the difference! Only in a couple of places is it (which is the best/most correct) done that way (lines 241 or 311).

6. A short paragraph about the study's limitations must be written at the end of the discussion. For example, nowhere in the text is the choice of "doses" of sulforaphane explained. Only in this part of the text do the doses in the two quoted studies "shyly" slip through (given in parentheses)! For example, 0.5 g/kg of SFN would, by simple analogy, mean ten grams for the human body, and no "medicine/drug" is given in those quantities.

7. Please also check the correctness of the citations for the mentioned studies. I am not sure that in MDPI journals, the page number is given as "p." and then the corresponding “number(s)!”

Author Response

  1. Please introduce the meaning of the acronyms used after the FIRST mention (both in the summary/abstract and in the text of the paper itself, starting with the Introductory part), e.g. (abstract) HFD (OK, you can tell from the title), but DIO (no clue for readers).

-Thanks for the suggestion. We have made edits accordingly. (page 1, lines 20 and 25)

  1. Check whether all Latin words/expressions (via, in vitro, etc.) are in italics throughout the text. I believe they are not!

-Ok, apologies. We have made edits accordingly.

  1. The tested bioactive compound (sulforaphane) is not well known to the general readership (except perhaps as a neoplastic agent). Therefore, it would be helpful to include its chemical structure (possibly with the formula of glucoraphanin, its glucosinolate precursor) in the text, as a separate figure, or as part of some figure with the study results.

-Thanks for the suggestion. We originally prepared a graphical abstract with the chemical structure of sulforaphane but since Foods does not require a graphical abstract, we have added the structures in Figure 1.

  1. Some parts of the text with experimental protocols must be completed, specifically section 2.2.1. and 2.2.2. In the specified order, the fluorescences were measured, and/or the samples were placed in the scintillation counter, and then/so what? Please complete, i.e., clarify!

-Thanks for the suggestion. We have made edits as suggested. (page 4, lines 172-174, 185-189)

  1. Put everywhere with the results of the study, along with claims that the observed differences (for the measured parameters) are significant (or a "strong" effect was obtained), at what level (give P value!), as well as by how much (is a specific parameter higher/lower) in percentages. Namely, the results are (mostly) presented through histograms, from which it takes work to see the difference! Only in a couple of places is it (which is the best/most correct) done that way (lines 241 or 311).

-Thanks for the instructive suggestion. The manuscript was edited with the result section changed as suggested.

  1. A short paragraph about the study's limitations must be written at the end of the discussion. For example, nowhere in the text is the choice of "doses" of sulforaphane explained. Only in this part of the text do the doses in the two quoted studies "shyly" slip through (given in parentheses)! For example, 0.5 g/kg of SFN would, by simple analogy, mean ten grams for the human body, and no "medicine/drug" is given in those quantities.

-Thank you for the comment. Sorry for the confusions but in our first two dietary intervention studies, the concentration of SFN 0.5 g/kg and 0.25 g/kg are not the proportion to body weight. It’s the diet concentration (0.5 g and 0.25 g SFN/kg diet). Based on our record, mouse ate about 2.1~2.5 g per day, meaning SFN they consumed is about 0.5~1.25 mg per day. Therefore, the animal dose of our first experiment is about 40 mg/kg as calculated based on a body weight of 30 g. Thus, we designed the oral gavage study be 40 mg/kg bw. The human equivalent dosage translated would be 200 mg for human weigh 60 kg, which is calculated as instructed https://doi.org/10.1096/fj.07-9574LSF, and https://www.ncbi.nlm.nih.gov/pmc/articles/PMC4804402/. The bioavailable sulforaphane from broccoli ranged 1.89-3.7 mg/g (https://doi.org/10.3390/foods10081927), meaning that human need to consume about 1.1 kg of broccoli to get the dose we tested in mouse. Therefore, in the second study, we lowered the concentration by half and shortened the experiment duration. As a commercialized product, the concentration of sulforaphane supplement ranges from 20 mg to 150 mg per capsule for human. Therefore, our findings are physiologically relevant since the dosage we tested are achievable by SFN supplement. We have added this to the discussion section (page 11, lines 395-403).

  1. Please also check the correctness of the citations for the mentioned studies. I am not sure that in MDPI journals, the page number is given as "p." and then the corresponding “number(s)!”

-Thank you for the valuable comments. We used the Endnote style “Numbered” for the references. The appropriate reference style Chicago-MDPI has been used now.

Reviewer 2 Report

Comments and Suggestions for Authors

Dear author, 

The manuscript entitled ''Sulforaphane ameliorates high-fat diet-induced metabolic abnormalities of young and middle-aged obese male mice'' was reviewed. After carefully reviewing the above mentioned article, I found there are some issues to be addressed, including : 

- Abstract section: Authors are invited to mention the main obtained results 

- The introduction section should be developed properly and cite at the end the originality and novelty of the study. 

- Authors are invited to add the ethical approval code

- The results are well presented but poorly discussed 

Author Response

- Abstract section: Authors are invited to mention the main obtained results

Response: Thank you for the suggestion. In the Abstract from line 20 to line 27, we have illustrated our major findings from animal studies and cell treatment experiment. Specific details were added as suggested.

- The introduction section should be developed properly and cite at the end the originality and novelty of the study.

Response: Thank you for the instructive suggestion. We have made edits as suggested (Page 2, lines 69-73).

- Authors are invited to add the ethical approval code

Response: Apologies for the inconvenience. Approval code has been added.

- The results are well presented but poorly discussed

Response: Thank you for the comment. In the discussion section, we have discussed the reason for the different phenotypes we observed in long- and short-term intervention studies. We speculated that long-term high fat feeding interrupted pancreatic β-cells therefore dietary SFN showed little effects on glycemic control by the end of the long-term experiment (lines 357-362). We have also compared our results to other research with consistent findings that SFN supplement would enhance energy metabolism (lines 375-382).

Reviewer 3 Report

Comments and Suggestions for Authors

This manuscript aims to determine whether sulforaphane, a natural compound derived from cruciferous vegetables, can prevent or treat obesity and insulin resistance in mouse 18 models.

Please consider the following suggestions:

Abstract: Please provide all the full names of the abbreviations you used.

Line 25: Please use italic font for “in vitro”

I suggest you extend the conclusions section by using more data from your study.

Since you used many abbreviations, I suggest you provide a table of abbreviations at the end of your manuscript.

Comments on the Quality of English Language

Minor editing of English language required

Author Response

Abstract: Please provide all the full names of the abbreviations you used.

Response: Thank you for the instructive comment. We have made changes as suggested.

Line 25: Please use italic font for “in vitro”

Response: Apologies. We have made edits as suggested.

I suggest you extend the conclusions section by using more data from your study.

Response: Thank you for the instructive suggestion. We have added more details of our result in the conclusion section (page 12, lines 434-438).

Since you used many abbreviations, I suggest you provide a table of abbreviations at the end of your manuscript.

Response: Thanks for the suggestions. We have made adjustment accordingly (page 12, lines 449-452).